# Investigating Subjective Motor Activity Perception and Gait in Parkinson's Disease

Syrine Slim[1]    Arne Küderle[1]    Hamid Moradi[1]    Mehul Mittal[1]
Dr. Emmanuelle Salin[1]    Prof. Dr. med. Jürgen Winkler[2]    Prof. Dr. Björn M. Eskofier[1]
[1] *Machine Learning and Data Analytics Lab, Friedrich-Alexander-University Erlangen-Nürnberg, Germany*
[2] *Molecular Neurology Department, University Clinic Erlangen, Germany*

*Abstract*—Understanding the relationship between patient-reported symptom perception and objective measurements of gait in Parkinson's Disease (PD) can support and inform clinical decision-making. Inertial Measurement Units (IMUs) facilitate the measurement of objective gait parameters in free-living conditions and allow for longitudinal monitoring. In this work, 15 stride-level gait parameters and stride count, derived from foot-worn IMUs, were statistically evaluated in their relation to subjective patient perception of daily motor-activity levels. This study covered a duration of 186 ± 5.5 days in free-living conditions for 33 PD patients. The comparison between the perceived low and high motor activity levels showed maintained significance after Bonferroni correction and medium to large effect sizes (ranging between 1.24 and 0.4) for the median over the recording period of stride count, turning angle, gait speed, stride length, gait clearance parameters (max. foot clearance, max. sensor lift, and max toe clearance). The effect sizes decreased when comparing neighboring motor activity levels, and the parameters showed a gradual improvement with higher activity levels. The results of this work show that daily subjective reports of motor activity mirror gait changes both in stride count and microparameters in PD patients as detected through foot-worn IMUs. This study provides groundwork for connecting subjective patient-reported feedback and free-living gait evaluations on the macro- (stride count), and micro-parameter level, as a first step towards including Patient Related Outcomes (PROs) within digital health systems for PD monitoring.

*Index Terms*—Subjective motor activity perception, Objective gait measurements, IMU, free-living monitoring, micro gait parameters, Stride count, Parkinson's disease, patient-centered care, PRO

## I. INTRODUCTION

Parkinson's disease (PD) significantly affects patients' mobility and quality of life [1]. Gait impairments are among its most common and disabling symptoms, often manifesting as reduced gait speed, shuffling of gait, and increased variability of gait [2]. Klucken et al. [3] emphasized the importance of real-world gait monitoring, as such information will allow for diagnostic and personalized therapy effect monitoring and adjustment. Moradi et al. [4] proved that the use of objective longitudinal gait data allowed for the detection of medication change effect in patients with PD.

This is achievable with the use of wearable sensors that have been proven to be objective measures of gait parameters in PD [5] and are practical and increasingly integrable within at-home monitoring systems [3]. At the same time, the perception of patient symptoms is an important piece of information in therapy assessment and monitoring of disease progression [6].

Recent Food and Drug Administration (FDA) guideline [7] highlights the value of patient-related outcomes (PROs) as key measures in digital health technologies. In their roadmap to implementing patient-centered digital outcomes in a longitudinal free-living monitoring paradigm, Espay et al. [8] emphasize the benefits of using an e-diary for patient subjective reporting to their symptoms, such as granular longitudinal changes in physical activity. [8], [9] recommend that future work explore how objective measures gathered by sensors align with end-points of interest, including patient input and mobility assessment.

Prior research has established links between subjective patient feedback and objective gait measurements in both lab and free-living settings. Leavy et al. [10] collected gait data over 7 days in both lab and free-living conditions, measuring step velocity, step length, gait variability in the lab, and stride count along with minutes of brisk walking (speed $> 1.05 m/s$) in daily life. Patients completed the patient-reported walking difficulties questionnaire (Walk-12G) once. Moderate correlations were found between perceived walking difficulties and objective measures such as gait speed, variability, and stride count. Mantri et al. [11] also compared patient-reported motor activity levels (questionnaire was filled once) with gait data collected over 7 days in free-living conditions. Step count moderately correlated with self-reported physical activity. In their work, Knudson et al. [12] examined the relationship between subjective reports of symptom impact on daily living and sensor-based measures of bradykinesia and dyskinesia over 6 days in free-living conditions. They found that both subjective and objective measures (step count and moderate to vigorous physical activity) significantly predicted Movement Disorder Society-Unified Parkinson's Disease Rating Scale Part II (MDS-UPDRS II) scores, supporting the value of integrating both perspectives for a fuller clinical picture.

While prior studies established meaningful links between subjective feedback and objective gait data, they were limited by infrequent patient reporting and short monitoring durations—typically 6 to 7 days with questionnaires completed only once. This restricts our understanding of day-to-day symptom fluctuations. Moreover, as defined by Del Din et al. [13], gait parameters can be categorized into macro- and micro-level. While macro-level parameters such as stride count have been examined in free-living conditions, micro-level parameters like gait speed, stride length, and stride time

have, to our knowledge, primarily been studied in laboratory settings, with limited investigation in free-living conditions. Additionally, micro-level gait dynamics—such as stride angles and gait clearance—remain underexplored, despite their potential to provide further insight into symptoms like shuffling of gait, postural instability [2], [5], and their usefulness in fall prediction [14].

In this work, we investigate whether daily subjective reports of motor activity reflect corresponding changes in gait parameters in individuals with PD, as captured through foot-worn inertial sensors. By establishing this link, we shed light on the unexplored relationship between perceived motor activity in free-living conditions and objective micro-level gait parameters, contributing to PRO oriented, holistic care of PD using digital health apps and wearable sensors.

## II. METHODS

### A. Data Acquisition

The data for this study were obtained from the ParkinsonGo study (ethical protocol No. 22-313-Bm). The inclusion criteria in this study were: age over 18 years, a diagnosis of idiopathic PD, Hoehn & Yahr stage (H&Y) I-IV, an MDS-UPDRS III score $\geq$ 13, a Minimal Status Test (MMST) score $\geq$ 26 indicating no significant cognitive impairment, the ability to walk at least 4x10 meters unaided, and the ability to operate a smartphone or tablet. The exclusion criteria were: patients with tremor-dominant PD without gait impairment or lower-limb bradykinesia, patients with a high frequency of falls ($>$1 fall/quarter), with severe OFF-related freezing, and the inability to walk independently indoors. A total of N=80 patients in accordance with these criteria were recruited from seven neurology clinical centers. Sensor-based measurements were obtained using foot-worn inertial measurement units (IMUs) under free-living conditions. Each participant was provided with the ParkinsonGo app on a smart device for completing questionnaires, along with two IMUs placed on the dorsum of each foot. The IMUs continuously recorded three-dimensional gyroscope and accelerometer data at a sampling frequency of 102.4 Hz. All used devices were supplied by Portabiles Healthcare Technologies. Participants were instructed to wear the sensors during daytime hours at least once a week and to complete a brief questionnaire at the end of each recording day, reporting their subjective perception of motor activity levels. The questionnaire asked: "Did you move a lot today?", with four response options: "I was very active" (high activity), "I was moderately active" (moderate activity), "I wasn't active" (low activity), and "My mobility was very restricted due to OFF phases and pain" (no activity). The questionnaire responses reflect the participants' subjective perception of their motor activity level on the respective recorded day. Due to the very limited number of 'no activity' recording days compared to the high, moderate, and low activity recordings, this label was excluded from the analysis. The questionnaires were developed in collaboration with Parkinson's disease specialists from the University Clinic of Erlangen and were designed to be patient-friendly and suitable for frequent completion,

and structured to mirror the types of questions typically asked during clinical visits.

### B. Data Handling and Pre-processing

Due to screening failures, 7 patients were excluded, resulting in an initial cohort of n=73. Due to data acquisition issues, n=51 patients completed the study, of whom n=47 both wore the sensors and submitted questionnaires. After filtering for patients who met the following criteria:

- Participated in the study for at least 100 days.
- Had full recording days within their participation period. For clarity, throughout this paper, a recording day is defined as a day with at least 1 minute of walking duration and a completed motor-activity questionnaire.
- Submitted each of the three motor activity level responses (low, moderate, and high) at least once during their participation.

N = 33 patients met the aforementioned criteria and were included in the final analysis. The main characteristics of the selected cohort are presented in Table I. The considered patients primarily had H&Y scores of I and II.

TABLE I: Patient characteristics (n=33). Values are mean $\pm$ standard deviation or counts for sex.

| Characteristics | Value |
|---|---|
| Sex (m/f) | 20/13 |
| Age [years] | $60.85 \pm 10.78$ |
| Height [cm] | $174.39 \pm 9.86$ |
| MDS-UPDRS III | $21.31 \pm 5.66$ |
| H&Y | $1.93 \pm 0.37$ |

TABLE II: Mean$\pm$SD of recording days and stride counts per activity level across 33 patients.

| Reported Motor Activity Level | Low Activity | Moderate Activity | High Activity |
|---|---|---|---|
| Number of recording days | $16.6 \pm 17.4$ | $43.4 \pm 28.5$ | $18.3 \pm 12.6$ |
| Stride count | $2083.9 \pm 1396.4$ | $3165.7 \pm 2065.3$ | $4694.5 \pm 2794.2$ |

Stride segmentation was performed on the raw gyroscope and accelerometer data for the left and right feet, and spatiotemporal stride-wise gait parameters were extracted. The gait-parameter extraction was performed using a custom algorithm. The extracted micro-gait parameters are: Gait Speed [m/s], Stride Length [cm], Stride Time [s], Swing Time [s], Stance Time [s] [15], Swing Time [%], Stance Time [%] (the relative durations of the swing and stance phases expressed as a percentage of the total stride time), Heel Strike Angle [°] (HS), Toe Off Angle [°] (TO), Max. Toe Clearance [cm] (TC), Max. Foot Clearance [cm], Max. Sensor Lift [cm] [16], Landing Impact [g], defined as the magnitude of the vertical acceleration around the heel strike [17], Max. Lateral Excursion [cm] (The maximum lateral deviation of the foot from an imaginary straight line connecting the start and end positions of a given stride), Turning Angle [°] (defined as the change in forward-facing orientation between the first and last sample of each stride, considering only the rotation around the vertical). We further computed the daily stride count,

considering only days with a total stride duration exceeding one minute, as a macro-parameter.

### C. Statistical Analysis

In this work, we aimed to compare gait parameters based on the patient-reported subjective motor activity level (low, moderate, and high) provided by the daily questionnaire. Due to the unsupervised nature of the data collection, recording days without completed questionnaires were excluded. Table II presents the average number of recording days per motor activity level across the selected cohort (n=33) and the corresponding stride count. The average duration of study participation across patients (duration between the first and last recording day) was $186.2 \pm 56.5$ days, the average number of recording days within this period was $88 \pm 53.4$ days.

Each recording day was labeled according to the corresponding motor activity level. For each parameter, descriptive statistics: mean, median, standard deviation, root mean square (RMS), coefficient of variation (CV), and the 5th and 95th percentiles were computed. We conducted two analyses:

*1. Full-period aggregation:* For each participant, gait parameter statistics were aggregated over the entire recording period, grouped by motor activity level. This resulted in one data point per level per participant. The analysis was then performed on each computed metric, covering both micro and macro gait parameters.

*2. Windowed aggregation:* In addition, a sliding window approach was applied using windows of one and two months, with no overlap between consecutive windows. Only windows encompassing the 3 different questionnaire answers were included. For each valid window, gait parameter statistics were computed separately for each motor activity level. The window was advanced sequentially across the timeline. This analysis allowed us to examine whether temporal granularity (1-month and 2-month windows vs. the full-recording period aggregation) influenced the relationship between gait characteristics and perceived activity level. Subsequently, a paired statistical analysis was conducted for both methods. For normally distributed data, the paired Student's t-test was applied; otherwise, a paired Wilcoxon test was used. Resulting p-values were corrected using the Bonferroni correction for each activity pair separately, and Hedges $g$ effect size was computed. This analysis was performed for the metrics of macro and micro-gait parameters.

## III. RESULTS

For simplicity, the results are summarized using two metrics: the median as a measure of central tendency, and the CV as a measure of variability. We report $p$-values before and after Bonferroni correction, and effect sizes using Hedges $g$. Generally, the mean showed similar results to the median metric, and the CV showed the most frequent maintained significance after correction compared to the other variability measures.

The $p$-value significance levels are annotated as follows: **ns**: $0.05 < p \leq 1.00$; **\***: $0.01 < p \leq 0.05$; **\*\***: $0.001 < p \leq$ 0.01; **\*\*\***: $p \leq 0.001$. To interpret the magnitude of effects, we adopted the following thresholds for Hedge' $g$: small (0.2–0.5), medium (0.5–0.8), and large ($>0.8$).

### A. Full-Period Aggregation Analysis

This section presents the results of the paired analysis comparing the 3 patient-reported motor activity levels across the full recording period.

*1) Macro-Parameter: Stride Count:* Stride count showed statistically significant differences between activity levels only in the median metric; it decreased with lower reported activity, with the largest effect observed in the comparison between the two extremes (low vs. high), yielding a Hedges g of $-1.14$. Comparisons between low vs. moderate and moderate vs. high yielded medium effect sizes of $-0.57$ and $-0.62$, respectively. Figure 1 illustrates the progressive increase in stride count medians across activity levels.

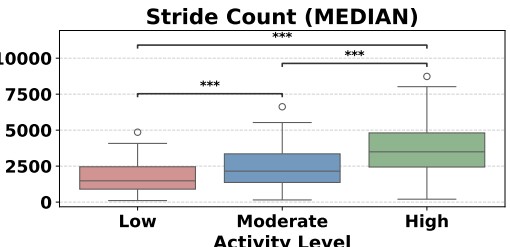

Fig. 1: Daily stride count across the three self-reported motor activity levels.

*2) Micro-Parameters:*

*a) Fundamental gait parameters:* Fig. 2 shows as an example the median gait speed and associated stride count per patient across the three reported activity levels (low, moderate, high). A clear visual separation is observed, with the low activity level (in red) almost consistently associated with lower values. Patient 8 showed higher gait speed on low activity days, with similar stride counts across all levels. Patient 21 walked less but faster on low activity days, while patient 15 showed the opposite pattern—more strides at slower speeds.

Table III presents the uncorrected and Bonferroni-corrected $p$-values, significance levels, and Hedges $g$ effect sizes for gait speed, stride length, stride time, and swing time percentage for each of the three motor activity level comparisons.

The moderate vs. high comparison yielded the smallest effect sizes and showed no significance after correction. In contrast, the low vs. high comparison demonstrated the largest effect sizes and the most retained significances after correction across all gait parameters. Median values of gait speed and stride length increased with activity level (reflected by negative effect sizes). Statistical significance was maintained in the low vs. high comparison for both parameters, with stride length also showing significance in the low vs. moderate comparison. Hedges' $g$ reached medium levels for low vs. high and small levels (0.2–0.4) for adjacent levels. The trend in median gait speed showing a gradual increase with activity level is visualized in Fig. 3-a. The progressive decrease of stride length

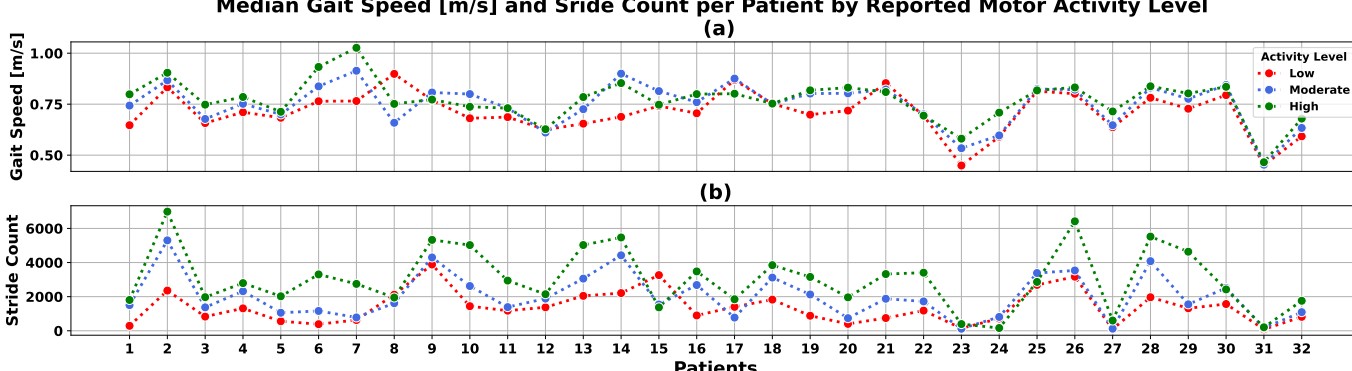

Fig. 2: Trend line: Median gait speed [m/s] **(a)** and stride count **(b)** per patient across the full recording period by reported activity level.

across reported motor activity levels is illustrated in Fig. 3-b. For CV, Gait speed and stride length showed comparable effect sizes, with stride length reaching statistical significance after correction in the low vs. high comparison. Gait speed CV showed the largest effect sizes across comparisons but did not reach significance after correction. Swing time percentage showed no significant differences in median values, but its CV (shown in Fig. 3) decreased with activity level, showing medium effect sizes and statistically significant differences when comparing the low activity level to both moderate and high. The same pattern was naturally observed for stance time percentage. Stride time showed small effect sizes and no statistically significant differences across comparisons.

*b) High fidelity gait parameters:* Table IV presents the medians of angular and spatial gait parameters across the three motor activity level comparisons.

The turning angle consistently decreased with higher activity levels and demonstrated the largest effect size of 1.24 for the low vs. high comparison. This strong effect was preserved across all activity level pairs and was accompanied by statistical significance after correction (** for neighboring levels, *** for extremes). The CV of the turning angle increased with higher activity levels and showed a similar significance pattern to the median.

Gait clearance parameters—namely, max. foot clearance, sensor lift, and TC—increased with motor activity level, showing medium effect sizes ranging from 0.4 to 0.5 in the low vs. high comparisons. Statistical significance after correction was maintained between the extreme levels and remained significant in the low vs. moderate comparison for max. TC and in the moderate vs. high comparison for max. sensor lift.

The HS angle also increased with higher activity levels, showing small effect sizes across the three comparisons and reaching statistical significance (after correction) only for the low vs. high pair. The TO angle showed slightly higher effect sizes than the HS angle, but the statistical significance did not hold after correction. Finally, max. lateral excursion exhibited negligible effect sizes and lacked statistical significance both before and after correction.

Among the CV results, only landing impact—aside from turning angle—showed statistical significance after correction ($p = 0.012$ (*)), accompanied by a small effect size of 0.38.

### B. Effect of Windowing on Activity-Level Discrimination

Table V presents the effect sizes (Hedges' $g$) for the median values of spatiotemporal gait parameters across aggregation methods, with parameters ordered from highest to lowest effect size in the full-period method. The temporal granularity introduced by the 1- and 2-month windowing approaches did not substantially affect the effect sizes relative to the full-period method. This consistency also extends to the low vs. moderate and moderate vs. high motor activity level comparisons (not shown in the table). Moreover, the ordering of parameters by effect size is largely preserved across the windowing methods compared to the full-period aggregation. Furthermore, statistical significance observed in the full-period analysis was retained in the windowed methods—whenever a parameter was significant after Bonferroni correction in the full-period analysis, it also reached significance in the corresponding windowed analyses. The corrected significances in the windowing methods were more frequent, but the effect sizes were comparable to the full-period method.

### IV. DISCUSSION

#### A. Full period aggregation results

Due to the large number of comparisons (122), the Bonferroni correction reduced the number of statistically significant findings. To address this, we complemented our analysis with Hedges' $g$ effect sizes. We suggest that parameters showing uncorrected significance alongside medium to large effect sizes be further validated in future studies.

The stride count outranked all micro parameters in effect size, except for the turning angle, and showed maintained significance after correction when comparing neighboring and extremes motor activity levels. The effect size also decreased granularly with the increase in activity levels. This confirms that patients interpreted the questionnaire meaningfully and related their responses to the perceived level of activity- they took more strides when they answered "I was very active".

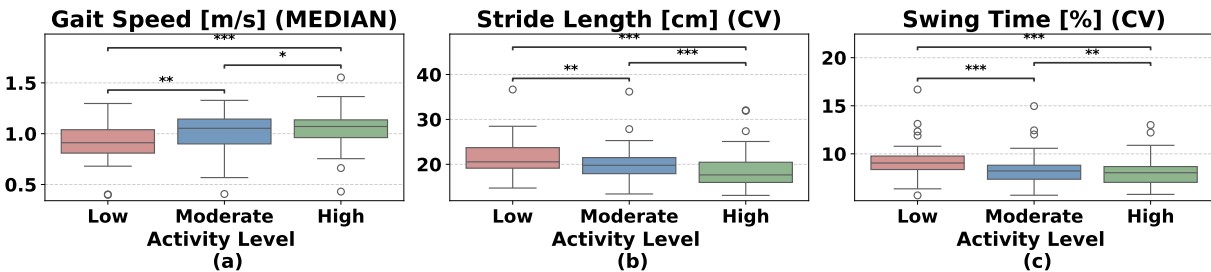

Fig. 3: Gradual progression of the median measure of Gait Speed (**a**), and the CV measure of Stride Length (**b**) and Swing Time percent (**c**) across the three self-reported motor activity levels with uncorrected p-value significances.

TABLE III: Comparison of fundamental gait parameters across motor activity level pairs, showing uncorrected and Bonferroni-corrected $p$-values (correction over 122 values), their mapped significance, and Hedges' $g$ effect sizes. The maintained significances after correction and their effect sizes are bolded.

| Parameter | Metric | Low vs High | | | Low vs Moderate | | | Moderate vs High | | |
|---|---|---|---|---|---|---|---|---|---|---|
| | | p (unc.) | p (Bonf.) | Hedges' g | p (unc.) | p (Bonf.) | Hedges' g | p (unc.) | p (Bonf.) | Hedges' g |
| Gait Speed [$m/s$] | Median | < 0.001 (***) | **0.006 (**)** | **-0.63** | 0.002 (**) | 0.20 (ns) | -0.42 | = 0.02 (*) | 1.00 (ns) | -0.19 |
| | CV | 0.001 (**) | 0.14 (ns) | **0.76** | 0.12 (ns) | 1.00 (ns) | 0.29 | < 0.001 (***) | 0.06 (ns) | 0.47 |
| Stride Length [$cm$] | Median | < 0.001 (***) | **0.001 (**)** | **-0.56** | < 0.001 (***) | **< 0.001 (***)** | **-0.37** | 0.006 (**) | 0.67 (ns) | -0.18 |
| | CV | < 0.001 (***) | **0.04 (*)** | **0.61** | 0.004 (**) | 0.41 (ns) | 0.28 | < 0.001 (***) | 0.005 (ns) | 0.35 |
| Stride Time [$s$] | Median | 0.008 (**) | 0.92 (ns) | 0.24 | 0.009 (**) | 1.00 (ns) | 0.18 | 0.30 (ns) | 1.00 (ns) | 0.06 |
| | CV | 0.03 (*) | 1.00 (ns) | 0.42 | 0.44 (ns) | 1.00 (ns) | 0.12 | 0.03 (*) | 1.00 (ns) | 0.30 |
| Swing Time [%] | Median | 0.008 (**) | 0.96 (ns) | -0.16 | 0.007 (**) | 0.77 (ns) | -0.12 | 0.42 (ns) | 1.00 (ns) | -0.03 |
| | CV | < 0.001 (***) | **0.01 (**)** | **0.53** | < 0.001 (***) | **0.002 (**)** | **0.31** | 0.004 (**) | 0.46 (ns) | 0.23 |

TABLE IV: Comparison of the **median metric** of the high-fidelity gait parameters across motor activity level pairs, showing uncorrected and Bonferroni corrected p-values (correction over 122 values) with the corresponding significance, and Hedges' $g$ effect sizes. The maintained significances after correction and their effect sizes are bolded.

| Parameter | Low vs High | | | Low vs Moderate | | | Moderate vs High | | |
|---|---|---|---|---|---|---|---|---|---|
| | p (unc.) | p (Bonf.) | Hedges' g | p (unc.) | p (Bonf.) | Hedges' g | p (unc.) | p (Bonf.) | Hedges' g |
| Turning Angle [$deg$] | < 0.001 (***) | **< 0.001 (***)** | **1.24** | < 0.001 (***) | **0.001 (**)** | **0.75** | < 0.001 (***) | **0.008 (**)** | **0.51** |
| Max. Foot Clearance [$cm$] | < 0.001 (***) | **0.008 (**)** | **-0.53** | 0.005 (**) | 0.563 (ns) | -0.29 | 0.002 (**) | 0.218 (ns) | -0.25 |
| Landing Impact [$g$] | < 0.001 (***) | **0.005 (**)** | **-0.49** | < 0.001 (***) | 0.056 (ns) | -0.32 | 0.040 (*) | 1 (ns) | -0.17 |
| Max. Sensor Lift [$cm$] | < 0.001 (***) | **0.001 (**)** | **-0.44** | 0.001 (**) | 0.169 (ns) | -0.21 | < 0.001 (***) | **0.023 (*)** | **-0.23** |
| Max. Toe Clearance [$cm$] | < 0.001 (***) | **< 0.001 (***)** | **-0.43** | < 0.001 (***) | **0.003 (**)** | **-0.31** | 0.068 (ns) | 1 (ns) | -0.10 |
| Heel Strike Angle [$deg$] | < 0.001 (***) | **0.033 (*)** | **-0.34** | 0.004 (**) | 0.408 (ns) | -0.20 | 0.130 (ns) | 1 (ns) | -0.13 |
| Toe Off Angle [$deg$] | 0.003 (**) | 0.292 (ns) | **0.40** | 0.038 (*) | 1 (ns) | 0.25 | 0.010 (**) | 1 (ns) | 0.15 |
| Max. Lateral Excursion [$cm$] | 0.762 (ns) | 1 (ns) | 0.03 | 0.951 (ns) | 1 (ns) | 0.01 | 0.613 (ns) | 1 (ns) | 0.03 |

That is, PD patients connect the perceived activity level with gait.

While most CV-based differences did not retain statistical significance after correction, the exceptions—stride length and stance time percentage—suggest a less consistent gait pattern on the stride-to-stride level in the perceived lower activity levels. Increased gait variability, particularly in stride time, has been linked to postural instability in PD [5]. It's also noteworthy that the stride time did not show any significance, and that the gait speed decrease in lower activities was driven primarily by the reduction in stride length. The significance of spatial parameters further underscores the value of specialized gait analysis systems (in this case, foot-worn IMUs) capable

of accurately capturing spatial metrics, allowing for reliable monitoring of PD patients.

Porta et al. [18] examined the relationship between objectively measured physical activity and gait parameters over a 3-month period- the longest monitoring that pervious work covered to our knowledge. They found correlations with cadence and stride length, but no significant association with gait speed. In contrast, our results revealed that while stride length median retained statistical significance between low and moderate activity levels, gait speed—despite not reaching significance due to the strict Bonferroni correction—exhibited the highest effect size among all assessed micro-gait parameters, including stride length. This substantial effect size suggests a meaningful

TABLE V: Comparison of **effect sizes** (Hedges' g) for the **median** of gait parameters between low and high motor activity levels across the three aggregation methods: 1-month window, 2-month window, and full-period.

| Parameter | Full Period | 2-Month Window | 1-Month Window |
|---|---|---|---|
| Turning Angle [deg] | 1.24 | 1.07 | 0.85 |
| Gait Speed [m/s] | -0.63 | -0.66 | -0.53 |
| Stride Length [cm] | -0.56 | -0.58 | -0.48 |
| Max. Foot Clearance [cm] | -0.53 | -0.53 | -0.43 |
| Landing Impact [g] | -0.49 | -0.51 | -0.41 |
| Max. Sensor Lift [cm] | -0.44 | -0.40 | -0.34 |
| Max. Toe Clearance [cm] | -0.43 | -0.41 | -0.37 |
| Toe Off Angle [deg] | 0.40 | 0.48 | 0.40 |
| Heel Strike Angle [deg] | -0.34 | -0.26 | -0.28 |
| Stride Time [s] | 0.24 | 0.33 | 0.26 |
| Swing Time [%] | -0.16 | -0.25 | -0.17 |
| Stance Time [%] | 0.16 | 0.25 | 0.17 |
| Max. Lateral Excursion [cm] | 0.03 | -0.01 | 0.01 |

relationship between gait speed and activity levels. Given its known association with bradykinesia [2], [19], gait speed warrants further investigation and validation in future studies, particularly in the context of long-term monitoring.

An interesting finding in this study is the high significance of the turning angle in relation to motor activity levels. These results, however, should be interpreted with caution. On one hand, our observations align with Mellone et al. [20], who reported that PD patients exhibit larger turning angles compared to healthy controls. This supports the idea that, within PD patients, lower motor activity may be associated with increased turning angles as a compensatory mechanism for dynamic postural instability. On the other hand, due to the way turning angle is defined in our study—calculated for every detected stride, including those during straight walking (see Section II-B)—the observed differences may partly or entirely reflect environmental context rather than motor symptoms. For example, on high activity days, patients are more likely to engage in longer, outdoor walks involving a higher proportion of straight-line strides. Conversely, low activity days may involve more indoor movement, where navigation around furniture or confined spaces necessitates frequent and sharper turns. Therefore, the observed decrease in median turning angle on high activity days might be attributed to contextual differences in walking environments, rather than a direct change in turning behavior or motor function. Future work should aim to refine stride classification by distinguishing between straight and turning strides—potentially using annotated data or algorithms like those described by Rehman et al. [21]—to better isolate turning-related motor strategies from environmental effects.

Foot lift parameters (maximum sensor lift and maximum foot clearance) rank just below turning angle among high-fidelity metrics and follow gait speed overall. HS angle increased while TO angle decreased in high activity days compared to low. This, along with reduced foot lift, was linked to gait shuffle in patients with PD [5]. The max. lateral excursion did not reflect the activity level, and according to

our work, should be excluded from future validations.

In Fig. 2 illustrates how self-reports can vary in interpretation. As described in III-A2a, some patients may walk less but faster (e.g., patient 21), while others take more strides at slower speeds (e.g., patient 15).

While overall trends are consistent, personal patterns may vary, supporting the need to combine subjective feedback with objective gait data and analyze gait parameters jointly to understand individual gait patterns and assess the patient's gait impairments, potentially supporting clinicians in defining fitting therapies.

Overall, the analyzed gait parameters differ between patient-reported low and high motor activity levels. The parameters most reflective of activity level differences, particularly between low and high levels—based on effect sizes, statistical significance, and data interpretation, were: gait speed, stride length, max. foot clearance, max. sensor lift, max. TC, and HS angle. We suggest that the TO angle be further investigated, as it demonstrated comparable effect sizes to these parameters.

### B. Effect of Windowing on Activity-Level Discrimination

Finally, increasing the temporal granularity through 1- and 2-month windowing did not substantially affect the observed effect sizes compared to the full-period results. As described in Section II-C, the windowing methods yielded more data points than the full-period approach, which explains the more frequent instances of statistical significance after correction. We propose that future studies apply a similar analysis over a longer monitoring duration, given that the average recording period in this work was 186.2 ± 56.5 days. The windowing approach was used to explore whether a patient's interpretation of activity levels is relative to their broader condition over time. For instance, after a bad month followed by a good one, a patient might report "moderate" activity for that period—contextualized within that specific phase. As Parkinson's progresses, this relativity may grow more pronounced, suggesting that longer-term monitoring (e.g., over multiple years) could offer deeper insights. We suggest this be examined in future work.

### C. Limitations and Future Directions

It is important to note that our results were aggregated from a cohort with an H&Y of I and II. These results might not apply to patients with advanced PD, as well as patients who utilize walking aids, as that was an exclusion criterion for our study. To assess generalizability, future studies should extend this analysis to individuals with more advanced PD. In addition, this analysis did not assess the consistency of effects across different patient subgroups. We suggest that future work consider stratifying the cohort based on dominant symptoms or adherence to system usage. It would be interesting to examine whether patient engagement with the digital system influences the consistency of their self-reports relative to objective gait measurements. Another limitation is the contextual variability inherent in free-living gait data. Factors such as walking environment (e.g., indoor vs. outdoor), physical setting, or

time of day could influence gait behavior and turning angles, potentially confounding the observed associations with activity levels. Moreover, the questionnaire was intentionally kept simple, aiming to capture general perceived daily motor activity and support long-term adherence. While activity is subjective, our findings show consistency with step count and stride-level gait metrics. However, the questionnaire has not yet been formally validated. Future work should develop symptom-specific formats to better capture perceived motor fluctuations, without compromising adherence by lengthy e-diaries. It would also be beneficial in future studies to include validated PRO measures as a benchmark for comparison-ideally in digital format, such as the PRO-PD scale [22], administered less frequently than the daily questionnaires to balance patient workload. Finally, while several gait parameters showed significant or large effect size differences across activity levels, the clinical relevance of these findings remains to be established. Further research is needed to determine whether such measures can inform disease monitoring or support therapeutic decision-making in clinical practice.

## V. Conclusion

This work investigated the relationship between patient-reported motor activity levels in free-living conditions and objective gait parameters derived from foot-worn IMUs. In particular, we found that stride length and foot clearance parameters differed between perceived motor activities. This highlights the importance of using accurate gait measurement systems in the context of clinical monitoring of PD patients. The observed alignment between objective gait data and subjective reports is the first step to including PROs in digital health systems for monitoring PD patients, supporting a holistic, patient-centered approach. Future work should explore the clinical relevance of these findings and better distinguish between effects driven by environmental context and those stemming from motor symptoms, to enable their use in PRO-oriented therapy assessment and adjustment.

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
