# OpenReview forum: "Investigating Subjective Motor Activity Perception and Gait in Parkinson‘s Disease"
_IEEE.org/EMBS/BHI/2025/Conference — BHI 2025_

### Official Review · Reviewer_4RDb · 2025-06-25
**The study presents interesting findings but is missing important details throughout the paper**

**Confidence:** 4
**Clarity Of Writing:** fair
**Clinical Significance:** good
**Methodological Novelty:** good
**Overall Rating:** 4
**Final Rating:** 7

**Experiments And Results:**

fair

**Questions For The Authors:**

- This study claims that completing questionnaires for only 6 to 7 days is insufficient. What would be a more appropriate duration for data collection? Are there any references to support this recommendation?
- The motivation for investigating the associations between objective gait metrics and subjective self-reports remains unclear. Are there references demonstrating the clinical relevance or necessity of this analysis in Parkinson’s disease (PD) assessment, treatment, or management? What are the potential benefits of this analysis for clinical research or practice?
- In the last paragraph of the introduction, the authors suggest that understanding the relationship between perceived motor activity in free-living conditions and objective micro-level gait parameters could contribute to more heuristic, patient-centered care for PD via digital health applications and wearable sensors. This claim is difficult to follow and requires further clarification, along with supporting references to substantiate the argument.
- The exclusion criteria are unclear. Please provide a clear explanation of the exclusion rules applied in this study.
- The questionnaire included only a single item: “Did you move a lot today?”. Why did the study focus exclusively on this question? Please justify this decision.
- Are there any references supporting the validity and reliability of the questionnaire used? Ideally, the questionnaire should have undergone validation to ensure its appropriateness for this context.
- The study should consider including other validated “Patient-Reported Outcome Measures” (PROMs) related to mobility or ambulatory function as benchmarks for comparison.
-  Is there any reference to support the criterion that “only the strides within walking bouts longer than one minute are analyzed”?
-  In Figure 2, some participants display higher gait speed despite reporting low motor activity levels. This discrepancy should be discussed.
-  How was the aggregation window length determined? Why was it not based on standard units like days or weeks?
-  In the discussion section, The paper states, “This change might be an indication of compensatory behavior caused by bradykinesia and postural instability, leading to altered and less uniform distributions of stance and swing times across strides.” Are there references to support this interpretation?
-  While the study focuses on the associations between subjective activity levels and objective gait metrics, it should first discuss the associations between subjective and objective activity levels more broadly.
-  Similarly, in the discussion section, the authors state, “The subjective questionnaire responses align with objective gait measurements and provide meaningful context for interpreting the data in the medical context, supporting a patient-centered holistic monitoring approach.” This conclusion is difficult to follow without further background or references. More supporting literature is needed to justify this claim.

**Strengths:**

a. This study includes a sufficient number of participants and an adequate data collection duration.

b. Appropriate statistical analyses are applied in this study.

c. A comprehensive set of gait parameters is included in the analysis.

**Summary Of The Paper:**

This study investigates the relationship between subjective perceptions of daily motor activity and objective gait parameters in individuals with Parkinson’s disease (PD) during free-living conditions. Using foot-worn inertial measurement units (IMUs), gait data were collected from 33 PD patients over an average of 186 days, alongside daily self-reports answering the question: “Did you move a lot today?” (with responses categorized as low, moderate, or high activity).

The analysis focused on both macro-level gait parameters (stride count) and micro-level gait parameters (e.g., gait speed, stride length, turning angle, foot clearance). Results demonstrated significant associations between higher self-reported activity levels and improved gait parameters, including higher stride count, faster gait speed, longer stride length, and greater foot clearance.

Additionally, the study introduced a sliding-window analysis (1-month and 2-month windows) to examine whether temporal aggregation affected these associations, finding consistent results across different window sizes.

**Weaknesses:**

1. The clinical motivation and rationale for this study are unclear.
2. Many aspects of the experimental setup and analysis lack sufficient references or justification.
3. The questionnaire used has not been validated for reliability and validity.

---

### Official Review · Reviewer_RhKT · 2025-07-05
**Revision 1**

**Confidence:** 3
**Clarity Of Writing:** good
**Clinical Significance:** fair
**Methodological Novelty:** fair
**Overall Rating:** 6

**Experiments And Results:**

fair

**Questions For The Authors:**

Apart from the limitations of the study (no context variables measured, limited classification of the perceived activity and non-stratification of PD patients), the abstract of the manuscript lacks quantitative performance results of the proposed method as well as some undefined acronyms.

**Strengths:**

1) A great number of extracted gait parameters and statistics.

**Summary Of The Paper:**

The paper proposes to investigate the relationship between patient-reported motor activity levels and objective gait parameters measured by  foot-worn IMUs in patients with Parkinson's disease (PD).

**Weaknesses:**

1) The study does not take into account contextual variable measurements, which may affect the reported results.

2) Limitation in the classification of the perceived activity by patients (with just 3 different levels)

3) Non-stratification of the different patient sub-groups in Parkinson's disease (PD).

---

### Official Review · Reviewer_RTe2 · 2025-07-15
**The paper has been well written, explaining the background and research gap in a clear way. The results have been reported and discussed clearly.**

**Confidence:** 4
**Clarity Of Writing:** good
**Clinical Significance:** good
**Methodological Novelty:** good
**Overall Rating:** 6

**Experiments And Results:**

good

**Questions For The Authors:**

1- How could the sampling frequency of IMUs affect the results?
2- What is the custom algorithm to perform gait parameter extraction?
3- The authors seem to only use median and CV. How about mean, standard deviation, and root mean square. Why were they calculated if they were not used at all?
4- Were the statistical tests one-tailed or two-tailed?
5- Can this study be used for other movement-related conditions, such as stroke, multiple sclerosis, etc?

**Strengths:**

1- The problem statement, background research, and motivation have been clearly written.
2- The tables and figures are visually appropriate.
3- The results are discussed comprehensively.

**Summary Of The Paper:**

In this paper, 15 stride-level gait parameters and stride count derived from foot-worn IMUs were statistically compared to subjective patient perception of daily motor-activity levels. The results demonstrate that daily subjective reports of motor activity accurately reflect gait changes, including stride count and microparameters, in PD patients as identified via foot-worn IMUs.

**Weaknesses:**

1- All the abbreviations should be defined the first time they appear in the paper, such as MDS-UPDRS.
2- Some statements need citations, for example:
While macro-level parameters such as stride count have been examined in free-living conditions, micro-level features like gait speed, stride length, and stride time have primarily been studied in laboratory settings, with limited investigation in free-living conditions.
3- Authors should explain notable names and characters used in the paper, such as Hoehn & Yahr in Table I or Nr in Table II.

---

### Official Review · Reviewer_a7hk · 2025-07-17
**Strong concordance between subjective motor activity reports and IMU-derived gait metrics in Parkinson’s Disease**

**Confidence:** 4
**Clarity Of Writing:** good
**Clinical Significance:** great
**Methodological Novelty:** good
**Overall Rating:** 7
**Final Rating:** 7

**Experiments And Results:**

great

**Questions For The Authors:**

1. Have you considered adding environment-aware classification (e.g., indoor vs. outdoor walking) to distinguish context-driven variation in gait parameters such as turning angle?
2. Could you elaborate on the potential for integrating symptom-specific self-report modules in future iterations of your system (e.g., targeting freezing of gait or dyskinesia)?
3. Was adherence (to both sensor wear and questionnaire completion) assessed over time? If so, did it influence the reliability of the subjective-objective alignment?
4. Are you planning to test the approach in patients with advanced PD or those using assistive devices?
5. Do you foresee clinical implementation pathways where subjective and IMU-derived data could inform real-time medication adjustment or cueing interventions?

**Strengths:**

1. Novel dataset: The longitudinal nature of the dataset (over ~6 months) and daily subjective self-reporting is a significant contribution, especially in the context of free-living monitoring.
2. Comprehensive analysis: The authors not only examine standard macro metrics but also delve into micro-gait parameters and their temporal granularity using windowing.
3. Statistical rigor: Use of Bonferroni correction alongside Hedges’ g effect sizes offers a nuanced view of both significance and effect magnitude.
4. Clinical relevance: The work supports the feasibility of integrating digital health tools into daily PD monitoring for personalized care.
5. Interpretive insights: The discussion carefully considers environmental confounds (e.g., indoor vs. outdoor walking impacting turning angle), strengthening the reliability of conclusions.

**Summary Of The Paper:**

This paper explores the relationship between self-reported daily motor activity levels and objective gait parameters captured using foot-worn inertial measurement units (IMUs) in people with Parkinson’s Disease (PD). Over an average of 186 days of free-living monitoring, the authors analyzed macro- (e.g., stride count) and micro-level gait parameters (e.g., gait speed, stride length, turning angle, gait clearance) across different subjective activity levels (low, moderate, high). Statistical comparisons with Bonferroni correction and Hedges’ g effect sizes revealed significant correlations between subjective reports and objective metrics - particularly in turning angle and gait clearance features. The study emphasizes the utility of integrating subjective and objective measures for long-term, patient-centered PD monitoring and therapy planning.

**Weaknesses:**

1. Limited generalizability: The cohort includes only early-stage PD patients (H&Y 1–2) and excludes those with walking aids or OFF freezing episodes, which may limit the applicability to broader PD populations.
2. Environmental variability: While discussed, the influence of context (e.g., indoor vs. outdoor) on parameters like turning angle could be further mitigated with environment-aware segmentation.
3. Simplified questionnaire: The daily self-report item is too coarse to isolate specific symptom dimensions like bradykinesia, rigidity, or fatigue. A multi-item format could yield more diagnostic utility.
4. Lack of subgroup analysis: No stratification is performed based on age, medication regimen, or adherence patterns - this could provide further insight into symptom perception variability.
5. No validation with clinical scales: It would have strengthened the claims if authors had correlated subjective reports or IMU-derived features with clinician-rated scales like MDS-UPDRS Part II or III over time.

---

### Official Review · Reviewer_4HUN · 2025-07-21
**Connecting subjective patient-reported perception and objective gait assessment**

**Confidence:** 5
**Clarity Of Writing:** excellent
**Clinical Significance:** good
**Methodological Novelty:** good
**Overall Rating:** 7
**Final Rating:** 7

**Experiments And Results:**

great

**Questions For The Authors:**

Questions for the authors:
1.	How can you disentangle gait features that reflect motor function from those that reflect environmental context (e.g., turning angle)?
2.	Do results generalize to patients with more advanced PD or those with gait impairments (e.g., freezing of gait, falls)?
3.	Can the system detect longitudinal changes or deterioration in motor function over time?
Although this study focuses on cross-sectional associations, your long recording period offers a unique opportunity to explore predictive value for symptom progression. Addressing this could increase the impact of your work.

**Strengths:**

Main strenghts of the study can be summarized as follows:
•	This study reports a very good longitudinal experimental protocol with long free-living monitoring durations (≈186 days) in PD
•	The analysis bridges patient-reported motor experience (subjective evaluation) and sensor-derived gait metrics, (objective assessment) offering a holistic view aligned with real-world clinical needs.
•	The methodology is robust and reliable, with multiple aggregation levels (full period, 1- and 2-month windows), multiple statistical measures (median, CV, RMS), and corrections (Bonferroni). Even when statistical significance was reduced by correction, medium to large Hedges’ g values highlight meaningful effects.
•	The app-based questionnaire is accessible, allowing a Patient-Centered Data Collection, promoting compliance and real-world feasibility

**Summary Of The Paper:**

This paper examines the relationship between subjective daily self-reports of motor activity and objective gait parameters captured via foot-worn inertial measurement units (IMUs) in individuals with Parkinson’s Disease (PD) under free-living conditions. Thirty-three PD patients were monitored for an average of 186 days, during which they recorded their perceived daily activity levels using a mobile app (low, moderate, high), and wore IMUs on both feet. The authors analyzed macro (stride count) and micro gait features (e.g., stride length, gait speed, toe clearance, turning angle) across self-reported activity levels. Statistical comparisons (paired tests, Bonferroni correction, Hedges’ g) were performed across three aggregation strategies: full-period, 1-month windows, and 2-month windows. Results showed significant and progressive trends between subjective activity and objective gait parameters. Parameters such as turning angle, gait speed, and foot clearance were most sensitive to perceived motor activity. The paper supports the potential of integrating subjective perception and objective gait metrics in long-term, patient-centered PD monitoring

**Weaknesses:**

Some weaknesses can be found in the study and summarized as follows:
•	The cohort was composed of patients with mild PD symptoms (H&Y stages 1–2), limiting insights for moderate to advanced cases or those using walking aids, with limited generalizability of the model.
•	Some results (e.g., turning angle) may reflect contextual factors (indoor vs. outdoor walking) rather than intrinsic motor ability. This raises questions about interpretability of certain parameters.
•	While simplicity supports adherence, the lack of symptom-specific questions (e.g., freezing, fatigue) reduces the resolution of subjective reporting.
•	The subjective reports were not validated against clinical ratings (e.g., MDS-UPDRS Part II/IV), and the clinical utility of the findings remains unquantified.
•	Potential heterogeneity (e.g., tremor-dominant vs. akinetic-rigid phenotypes) was not explored, which could influence gait perception and reporting.
•	While understandable due to low counts, the omission of days with severe symptoms or OFF phases (i.e., exclusion of "no activity" days) may miss clinically important extremes.